# Income disparities in COVID-19 vaccine and booster uptake in the United States: An analysis of cross-sectional data from the Medical Expenditure Panel Survey

Kenechukwu C. Ben-Umeh[1], Jaewhan Kim[2]*

1 Department of Pharmacotherapy, University of Utah College of Pharmacy, Salt Lake City, Utah, United States of America, 2 Department of Physical Therapy, University of Utah, Salt Lake City, Utah, United States of America

* Jaewhan.kim@utah.edu

## Abstract

COVID-19 vaccination has significantly decreased morbidity, hospitalizations, and death during the pandemic. However, disparities in vaccination uptake threatens to stymie the progress made in safeguarding the health of Americans. Using a nationally representative adult ($\geq$18 years old) sample from the 2021 Medical Expenditure Panel Survey (MEPS), we aimed to explore disparities in COVID-19 vaccine and booster uptake by income levels. To reflect the nature of the survey, a weighted logistic regression analysis was used to explore factors associated with COVID-19 vaccine and booster uptake. A total of 241,645,704 (unweighted n = 21,554) adults were included in the analysis. Average (SD) age of the population was 49 (18) years old, and 51% were female. There were disparities in COVID-19 vaccine and booster uptake by income groups. All other income groups were less likely to receive COVID-19 vaccines and booster shot than those in the high-income group. Those in the poor income group had 55% lower odds of being vaccinated for COVID-19 (aOR = 0.45, p<0.01). Considering the female population only, women with lower incomes may have greater disparities in access to COVID-19 vaccines than do males with lower incomes. Disparities in COVID-19 vaccination by income may have even greater implications as the updated vaccines are rolled out in the US without the government covering the cost as before.

## Introduction

With over 103 million confirmed cases of COVID-19 and more than 1.14 million deaths in the US as a result of COVID-19, the pandemic has posed and still poses a great threat to the public health of the US [1, 2]. The approval and recommendation of various COVID-19 vaccines tremendously decreased morbidity, hospitalizations and death as a result of the virus in the US [3]. At the early stage of the COVID-19 vaccine distribution, dissemination was initially intended for the extremely susceptible population such as long-term care residents, those with

**Data Availability Statement:** The data underlying the results presented in the study are available from the following DOI: 10.6084/m9.figshare.25043915.

**Funding:** The author(s) received no specific funding for this work.

**Competing interests:** The authors have declared that no competing interests exist.

numerous comorbidities and frontline healthcare workers. However, by April 2021 all adults in the US were eligible to be vaccinated irrespective of age, risk level, profession or state-specific prioritization rules [4]. Despite the fact that these vaccines have been made available to everyone willing to take them at no cost, concerns about equitable distribution of the vaccines still persists in the US [5].

Evidence from previous studies show that racial and ethnic minorities in the US have an increased risk of contracting COVID-19, having complications and dying as a result of the virus [6, 7]. Several studies have reported that disparities in COVID-19 vaccination uptake could exist across social determinants of health (SDOH) such as race and ethnicity, age, sex, and income levels [8–11]. These disparities in uptake have also been reported in the vaccination for other diseases such as influenza and pneumococcal virus and have been tied to an as intricate interactions of access to healthcare, distrust in the health system due to ongoing or previous discrimination and injustices, attitudes and general literacy [4, 12, 13].

Many prior studies in the US have focused on racial and ethnic disparities in COVID-19 vaccines uptake. Among people who were prioritized and received COVID-19 vaccine within the first month it was rolled out, over 60% were non-Hispanic White [14]. The second-dose initiation and completion rates in the US varied by income, race and ethnicity among other factors such as age, occupation and insurance which could lead to different protection levels in the population [4]. A recent study using CVS Health and Walgreens pharmacy data linked to Medicare claims found that racial and ethnic disparities still persisted for COVID-19 booster vaccinations among the elderly in the US [15].

These studies have explored disparities in either a single dose of the vaccine, initiation, and completion of the initial 2 doses or booster vaccinations independently. However, few studies have provided a comprehensive update on the issue of disparities in COVID-19 vaccine uptake using recent national survey data. Our study aimed to explore the presence of disparities in COVID-19 vaccine and booster uptake in the US and factors associated with vaccine uptake among those vaccinated in 2021 using the latest data from a nationally representative sample.

## Materials and methods

### Data

Our study follows Strengthening the Reporting of Observational Studies in Epidemiology (STROBE) reporting guideline. This study made use of the Medical Expenditure Panel Survey (MEPS) 2021 Full Year Consolidated File. The MEPS polls individuals, families, and healthcare professionals in the US to gather data on demographics, health conditions, health care use and spending, insurance status, access to care, employment status, and family relations from a sample of the civilian non-institutionalized population that is nationally representative. Data for all occupants of a household was submitted by a single household respondent or a group of household members. The survey subjects self-report information on their demographics (such as age, sex, and race/ethnicity), household income level, family relationships, health conditions (such as hypertension, diabetes, asthma, cancer), and use of healthcare, while information on health care spending is gathered from both household respondents and providers [16–18]. The 2021 Full Year Population Characteristics File included several new questionnaires including taking COVID-19 vaccine and booster shot and social determinant of health (SDOH) such as affordable housing, discrimination, and food insecurity. In the 2021 survey, the response rates varied between 60% to 87.6% based on the four data collection panels (referred to as Panel 23 to Panel 26) [19]. The MEPS has been conducted since 1996, and numerous research studies have utilized this data; hence its reliability and validity have been established by previous publications [20–22].

## Subjects

All adult respondents ($\geq$18 years old) in the survey were included. Those who had missing values in the outcomes and the covariates were excluded from the analysis.

## Outcomes

There are two questionnaires in the survey that asked participants if they ever had COVID-19 vaccine and if they ever had COVID-19 booster shot. Using these two questionnaires, binary outcomes were created (Yes-took COVID-19 vaccine, No-did not take COVID-19 vaccine; Yes-took COVID-19 booster shot, No-did not take COVID-19 booster shot).

## Covariates

The main independent variable was income level measured in the survey. The survey has family income as percent of poverty line in 2021 with 5 categories: poor (less than 100% of the poverty line in 2021), near poor (100%-125%), low income (125%-200%), middle income (200%-400%), and high income ($\geq$400%). In the analysis, the high-income group was the reference group. Age (18–30, 31–40, 41–50, 51–64, 65–74 and $\geq$75 years old in 2021), sex (male/female), race/ethnicity (non-Hispanic white, Hispanic, non-Hispanic black, non-Hispanic Asian, non-Hispanic other race), region (northeast, Midwest, south, west), type of insurance (private, Medicaid, Medicare, no insurance), marital status (married, widowed, divorced, separated, never married), high school graduate (Yes/No), US born (Yes/No), hypertension (Yes/No), diabetes (Yes/No), asthma (Yes/No), any cancer diagnosis (Yes/No), total number of office visits in 2021, and whether a subject had a usual source of care provider were also adjusted for in the regression. Office visit was defined as care provided in a doctor's office, group practice office, medical clinic, community health center, urgent care center, or school clinic [23].

## Statistical approach

Summary statistics such as mean, standard deviation (SD), and percent were used to summarize the population in the study. Because there were 5 groups of subjects by the income level, pairwise comparisons in the control variables were conducted. Weighted logistic regression was used to estimate odds ratio and p-values of the controlled variables because the outcome variables were binary outcomes. Taylor-series linearization method was used to calculate correct standard errors of the variables to reflect a complex sample design [24]. P-values less than 0.05 was defined as statistically significant. All analysis was conducted using STATA software version 18.0.

# Results

A total of 256,165,119 (unweighted n = 22,587) adult subjects were selected. Among them, 14,519,415 (unweighted n = 1,033) subjects were excluded because they had missing information such as education, hypertension, diabetes, cancer diagnosis, and usual source of care provider variables. The final population size in the analysis was 241,645,704 (unweighted n = 21,554). In the population, 10%, 4%, 11%, 28% and 47% of subjects were in the poor, near poor, low income, middle income, and high-income categories, respectively. Average (SD) age of the population in the survey was 49 (18) years, and 51% of the population were female. Across the income level, the sociodemographic characteristics of the population were different. About 62% of the survey population were non-Hispanic white while 12% were non-Hispanic black. A greater proportion of the non-Hispanic white population (71%) were in the high-income group compared to other races/ethnicity. Table 1 presents the weighted descriptive statistics for the study population by income categories.

**Table 1. Characteristics of population by income level.**

| | Family income as % of poverty line | | | | | | |
| | POOR | NEAR POOR | LOW INCOME | MIDDLE INCOME | HIGH INCOME | Total | p-value |
| | mean(SD)/n(%) | mean(SD)/n(%) | mean(SD)/n(%) | mean(SD)/n(%) | mean(SD)/n(%) | mean(SD)/n(%) | |
|---|---|---|---|---|---|---|---|
| N | 24,948,003 (10.3%) | 8,756,866 (3.6%) | 27,859,971 (11.5%) | 67,814,872 (28.1%) | 112,265,991 (46.5%) | 241,645,704 (100.0%) | |
| Age (continuous) | 47.254 (19.129) | 51.257 (20.067) | 49.232 (19.937) | 47.686 (18.362) | 49.328 (17.138) | 48.712 (18.171) | <0.001 |
| Age category | | | | | | | |
| 18–30 | 6,558,866 (26.3%) | 1,732,304 (19.8%) | 6,425,323 (23.1%) | 15,225,742 (22.5%) | 19,159,929 (17.1%) | 49,102,164 (20.3%) | <0.001 |
| 31–40 | 4,260,501 (17.1%) | 1,305,246 (14.9%) | 4,917,260 (17.6%) | 12,363,276 (18.2%) | 19,468,841 (17.3%) | 42,315,124 (17.5%) | |
| 41–50 | 3,155,405 (12.6%) | 1,152,771 (13.2%) | 3,689,939 (13.2%) | 11,124,077 (16.4%) | 19,468,781 (17.3%) | 38,590,972 (16.0%) | |
| 51–64 | 5,387,166 (21.6%) | 1,967,518 (22.5%) | 4,997,228 (17.9%) | 14,228,804 (21.0%) | 29,981,814 (26.7%) | 56,562,530 (23.4%) | |
| 65–74 | 3,196,731 (12.8%) | 1,334,356 (15.2%) | 4,021,566 (14.4%) | 8,800,663 (13.0%) | 15,423,646 (13.7%) | 32,776,963 (13.6%) | |
| ≥75 | 2,389,333 (9.6%) | 1,264,671 (14.4%) | 3,808,655 (13.7%) | 6,072,311 (9.0%) | 8,762,980 (7.8%) | 22,297,951 (9.2%) | |
| Sex | | | | | | | |
| Male | 10,597,306 (42.5%) | 3,653,173 (41.7%) | 12,712,256 (45.6%) | 32,814,766 (48.4%) | 57,807,735 (51.5%) | 117,585,236 (48.7%) | <0.001 |
| Female | 14,350,698 (57.5%) | 5,103,693 (58.3%) | 15,147,714 (54.4%) | 35,000,107 (51.6%) | 54,458,256 (48.5%) | 124,060,468 (51.3%) | |
| Race/Ethnicity | | | | | | | |
| Hispanic | 5,851,571 (23.5%) | 2,107,084 (24.1%) | 6,566,555 (23.6%) | 13,651,037 (20.1%) | 12,548,549 (11.2%) | 40,724,796 (16.9%) | <0.001 |
| Non-Hispanic White | 11,748,044 (47.1%) | 4,552,038 (52.0%) | 14,548,270 (52.2%) | 39,649,648 (58.5%) | 79,549,342 (70.9%) | 150,047,342 (62.1%) | |
| Non-Hispanic Black | 4,913,662 (19.7%) | 1,498,815 (17.1%) | 4,107,560 (14.7%) | 9,302,268 (13.7%) | 8,611,621 (7.7%) | 28,433,927 (11.8%) | |
| Non-Hispanic Asian | 1,411,016 (5.7%) | 285,352 (3.3%) | 1,574,771 (5.7%) | 2,772,217 (4.1%) | 8,726,020 (7.8%) | 14,769,376 (6.1%) | |
| Non-Hispanic other race | 1,023,709 (4.1%) | 313,576 (3.6%) | 1,062,815 (3.8%) | 2,439,702 (3.6%) | 2,830,459 (2.5%) | 7,670,262 (3.2%) | |
| Census region | | | | | | | |
| Northeast | 3,950,449 (15.9%) | 1,300,523 (14.9%) | 4,518,279 (16.2%) | 10,871,397 (16.0%) | 20,980,401 (18.7%) | 41,621,049 (17.2%) | <0.001 |
| Midwest | 4,705,515 (18.9%) | 1,530,701 (17.5%) | 5,697,269 (20.5%) | 13,697,557 (20.2%) | 25,291,322 (22.5%) | 50,922,364 (21.1%) | |
| South | 10,312,759 (41.4%) | 3,707,390 (42.3%) | 11,885,023 (42.7%) | 27,840,098 (41.1%) | 37,982,805 (33.9%) | 91,728,075 (38.0%) | |
| West | 5,932,234 (23.8%) | 2,218,252 (25.3%) | 5,718,277 (20.6%) | 15,393,179 (22.7%) | 27,947,232 (24.9%) | 57,209,174 (23.7%) | |
| Type of health insurance | | | | | | | |
| Private health insurance | 4,731,741 (19.0%) | 1,579,300 (18.0%) | 10,198,297 (36.6%) | 40,275,188 (59.4%) | 81,669,716 (72.7%) | 138,454,243 (57.3%) | <0.001 |
| Medicaid | 10,979,875 (44.0%) | 3,631,473 (41.5%) | 6,211,036 (22.3%) | 7,465,790 (11.0%) | 3,212,768 (2.9%) | 31,500,942 (13.0%) | |
| Medicare | 5,516,049 (22.1%) | 2,578,184 (29.4%) | 7,724,659 (27.7%) | 14,531,419 (21.4%) | 23,495,186 (20.9%) | 53,845,498 (22.3%) | |
| Uninsured | 3,720,338 (14.9%) | 967,908 (11.1%) | 3,725,979 (13.4%) | 5,542,475 (8.2%) | 3,888,321 (3.5%) | 17,845,022 (7.4%) | |
| Marital status | | | | | | | |
| Married | 6,518,141 (26.1%) | 2,722,510 (31.1%) | 11,028,055 (39.6%) | 32,291,130 (47.6%) | 73,249,374 (65.2%) | 125,809,210 (52.1%) | <0.001 |
| Widowed | 2,218,740 (8.9%) | 1,349,744 (15.4%) | 3,075,908 (11.0%) | 4,658,979 (6.9%) | 4,509,182 (4.0%) | 15,812,553 (6.5%) | |
| Divorced | 4,030,891 (16.2%) | 1,623,145 (18.5%) | 3,994,613 (14.3%) | 8,682,117 (12.8%) | 8,791,287 (7.8%) | 27,122,053 (11.2%) | |
| Separated | 1,159,395 (4.6%) | 183,397 (2.1%) | 826,202 (3.0%) | 1,211,983 (1.8%) | 706,031 (0.6%) | 4,087,008 (1.7%) | |
| Never married | 11,020,835 (44.2%) | 2,878,070 (32.9%) | 8,935,194 (32.1%) | 20,970,664 (30.9%) | 25,010,116 (22.3%) | 68,814,880 (28.5%) | |
| High school graduate | | | | | | | |
| No | 7,846,980 (31.5%) | 2,750,678 (31.4%) | 6,362,514 (22.8%) | 9,798,991 (14.4%) | 7,013,257 (6.2%) | 33,772,420 (14.0%) | <0.001 |
| Yes | 17,101,023 (68.5%) | 6,006,187 (68.6%) | 21,497,457 (77.2%) | 58,015,881 (85.6%) | 105,252,734 (93.8%) | 207,873,283 (86.0%) | |
| USA born | | | | | | | |
| No | 5,311,823 (21.3%) | 1,996,271 (22.8%) | 6,206,145 (22.3%) | 12,280,678 (18.1%) | 17,326,525 (15.4%) | 43,121,442 (17.8%) | <0.001 |
| Yes | 19,636,181 (78.7%) | 6,760,595 (77.2%) | 21,653,826 (77.7%) | 55,534,194 (81.9%) | 94,939,466 (84.6%) | 198,524,262 (82.2%) | |
| Hypertension | | | | | | | |
| No | 16,031,876 (64.3%) | 4,892,532 (55.9%) | 17,630,353 (63.3%) | 44,627,394 (65.8%) | 78,427,698 (69.9%) | 161,609,853 (66.9%) | <0.001 |
| Yes | 8,916,127 (35.7%) | 3,864,333 (44.1%) | 10,229,618 (36.7%) | 23,187,478 (34.2%) | 33,838,293 (30.1%) | 80,035,850 (33.1%) | |
| Diabetes | | | | | | | |
| No | 20,865,839 (83.6%) | 7,138,673 (81.5%) | 23,896,726 (85.8%) | 59,345,128 (87.5%) | 102,157,283 (91.0%) | 213,403,650 (88.3%) | <0.001 |

(*Continued*)

**Table 1.** (Continued)

| | POOR | NEAR POOR | LOW INCOME | MIDDLE INCOME | HIGH INCOME | Total | p-value |
|---|---|---|---|---|---|---|---|
| | mean(SD)/n(%) | mean(SD)/n(%) | mean(SD)/n(%) | mean(SD)/n(%) | mean(SD)/n(%) | mean(SD)/n(%) | |
| | | | | Family income as % of poverty line | | | |
| Yes | 4,082,164 (16.4%) | 1,618,193 (18.5%) | 3,963,244 (14.2%) | 8,469,745 (12.5%) | 10,108,708 (9.0%) | 28,242,054 (11.7%) | |
| Asthma | | | | | | | |
| No | 20,423,649 (81.9%) | 7,012,723 (80.1%) | 23,193,140 (83.2%) | 57,662,920 (85.0%) | 98,304,424 (87.6%) | 206,596,857 (85.5%) | <0.001 |
| Yes | 4,524,354 (18.1%) | 1,744,143 (19.9%) | 4,666,831 (16.8%) | 10,151,952 (15.0%) | 13,961,567 (12.4%) | 35,048,847 (14.5%) | |
| Any cancer | | | | | | | |
| No | 22,738,719 (91.1%) | 7,648,318 (87.3%) | 24,585,523 (88.2%) | 60,542,509 (89.3%) | 98,786,893 (88.0%) | 214,301,961 (88.7%) | 0.004 |
| Yes | 2,209,285 (8.9%) | 1,108,548 (12.7%) | 3,274,448 (11.8%) | 7,272,364 (10.7%) | 13,479,098 (12.0%) | 27,343,742 (11.3%) | |
| Number of office visits | 7.193 (16.601) | 7.969 (14.613) | 7.339 (13.138) | 7.253 (13.846) | 8.629 (13.410) | 7.922 (13.923) | <0.001 |
| Not English speaking | | | | | | | |
| No | 22,665,450 (90.9%) | 7,968,708 (91.0%) | 25,924,156 (93.1%) | 64,800,222 (95.6%) | 111,068,909 (98.9%) | 232,427,445 (96.2%) | <0.001 |
| Yes | 2,282,553 (9.1%) | 788,158 (9.0%) | 1,935,814 (6.9%) | 3,014,651 (4.4%) | 1,197,082 (1.1%) | 9,218,259 (3.8%) | |
| Usual source of care provider | | | | | | | |
| No | 9,107,381 (36.5%) | 2,342,941 (26.8%) | 8,768,074 (31.5%) | 20,863,867 (30.8%) | 27,794,803 (24.8%) | 68,877,065 (28.5%) | <0.001 |
| Yes | 15,840,622 (63.5%) | 6,413,925 (73.2%) | 19,091,897 (68.5%) | 46,951,006 (69.2%) | 84,471,188 (75.2%) | 172,768,638 (71.5%) | |
| Took COVID-19 vaccine | | | | | | | |
| No | 9,609,089 (38.5%) | 2,855,951 (32.6%) | 8,341,974 (29.9%) | 19,050,559 (28.1%) | 17,558,001 (15.6%) | 57,415,574 (23.8%) | <0.001 |
| Yes | 15,338,914 (61.5%) | 5,900,915 (67.4%) | 19,517,997 (70.1%) | 48,764,314 (71.9%) | 94,707,990 (84.4%) | 184,230,130 (76.2%) | |
| Took a booster shot | | | | | | | |
| No | 7,970,620 (47.1%) | 2,852,547 (44.9%) | 9,554,213 (46.3%) | 21,887,847 (42.2%) | 30,133,225 (30.9%) | 72,398,453 (37.4%) | <0.001 |
| Yes | 8,960,063 (52.9%) | 3,497,806 (55.1%) | 11,095,392 (53.7%) | 30,027,333 (57.8%) | 67,440,048 (69.1%) | 121,020,642 (62.6%) | |

Compared to those who were in the high-income group, all the other income groups were associated with lower odds of COVID-19 vaccine uptake. Those who were in the poor income group had 55% lower odds of being vaccinated for COVID-19 (aOR = 0.45, p<0.01), while those who were in the low-income group had 45% lower odds of being vaccinated for COVID-19 (aOR = 0.55, p<0.01). Older people were getting more COVID-19 vaccine compared to those aged 18 to 30 years. For example, the 65–74 years age group and the ≥75 years age group had 7 times (aOR = 7.33, p<0.01) and 10 times (aOR = 9.93, p<0.01) the higher odds of being vaccinated than the 18–30 years age group, respectively. Females had 17% higher odds of being vaccinated for COVID-19 than males (aOR = 1.17, p<0.01) (Table 2).

Similar results were found in the analysis using the male population only. Those who were in the poor income group, or the low-income group had 43% (aOR = 0.57, p<0.01) and 40% (aOR = 0.60, p<0.01) lower odds of being vaccinated for COVID-19 compared to those who were in the high-income group. Subjects who were covered by Medicaid (aOR = 0.58, p<0.01), Medicare (aOR = 0.45, P<0.01), or with no insurance (aOR = 0.35, p<0.01) had lower odds of receiving COVID-19 vaccine than those who were covered by private insurance (Table 3).

Considering the female population only, all other income groups compared to the high-income group had lower odds of being vaccinated for COVID-19. The poor income group had 64% lower odds of being vaccinated for COVID-19 (aOR = 0.36, p<0.01), while the low-income group had 58% lower odds (aOR = 0.49, p<0.01) compared to the high-income group. These numbers might indicate that women with lower incomes could experience more disparity in access to COVID-19 vaccines even more than men with lower incomes. Women who

**Table 2. Factors associated with taking a COVID-19 vaccine (all adults n = 241,645,704).**

| COVID-19 vaccine | Odds ratio | p-value | 95% CI | |
|---|---|---|---|---|
| Family income as % of poverty line | | | | |
| Poor | 0.45 | <0.01 | 0.37 | 0.55 |
| New poor | 0.51 | <0.01 | 0.39 | 0.66 |
| Low income | 0.55 | <0.01 | 0.46 | 0.66 |
| Middle income | 0.56 | <0.01 | 0.48 | 0.65 |
| High income | reference | | | |
| Age category | | | | |
| 18–30 | reference | | | |
| 31–40 | 1.08 | 0.36 | 0.92 | 1.26 |
| 41–50 | 1.32 | 0.01 | 1.08 | 1.60 |
| 51–64 | 2.11 | <0.01 | 1.76 | 2.53 |
| 65–74 | 7.33 | <0.01 | 3.32 | 16.19 |
| ≥75 | 9.93 | <0.01 | 4.29 | 23.02 |
| Female | 1.17 | <0.01 | 1.08 | 1.26 |
| Race/Ethnicity | | | | |
| Non-Hispanic White | reference | | | |
| Hispanic | 1.48 | <0.01 | 1.20 | 1.82 |
| Non-Hispanic Black | 1.07 | 0.48 | 0.89 | 1.28 |
| Non-Hispanic Asian | 3.29 | <0.01 | 2.14 | 5.06 |
| Non-Hispanic other race | 1.01 | 0.95 | 0.75 | 1.36 |
| Census region | | | | |
| Northeast | reference | | | |
| Midwest | 1.10 | 0.46 | 0.85 | 1.42 |
| South | 0.74 | <0.01 | 0.60 | 0.91 |
| West | 0.76 | 0.01 | 0.62 | 0.94 |
| Type of health insurance | | | | |
| Private health insurance | reference | | | |
| Medicaid | 0.55 | <0.01 | 0.47 | 0.64 |
| Medicare | 0.47 | <0.01 | 0.39 | 0.57 |
| Uninsured | 0.39 | 0.02 | 0.18 | 0.87 |
| Marital status | | | | |
| Married | reference | | | |
| Widowed | 0.78 | 0.03 | 0.63 | 0.97 |
| Divorced | 0.75 | <0.01 | 0.65 | 0.88 |
| Separated | 0.67 | <0.01 | 0.53 | 0.85 |
| Never married | 1.08 | 0.27 | 0.94 | 1.24 |
| High school graduate | 1.43 | <0.01 | 1.27 | 1.63 |
| USA born | 0.70 | <0.01 | 0.57 | 0.85 |
| Hypertension | 1.09 | 0.14 | 0.97 | 1.22 |
| Diabetes | 1.16 | 0.09 | 0.98 | 1.37 |
| Asthma | 1.14 | 0.05 | 1.00 | 1.29 |
| Any cancer | 1.08 | 0.38 | 0.91 | 1.29 |
| Number of office visits | 1.02 | <0.01 | 1.02 | 1.03 |
| Not English speaking | 0.99 | 0.95 | 0.74 | 1.33 |
| Usual source of care provider | 1.24 | <0.01 | 1.10 | 1.39 |

**Table 3. Factors associated with taking a COVID-19 vaccine (male only n = 117,585,236).**

| COVID-19 vaccine | Odds ratio | p-value | 95% CI | |
|---|---|---|---|---|
| Family income as % of poverty line | | | | |
| Poor | 0.57 | <0.01 | 0.44 | 0.73 |
| New poor | 0.61 | <0.01 | 0.44 | 0.85 |
| Low income | 0.60 | <0.01 | 0.47 | 0.76 |
| Middle income | 0.60 | <0.01 | 0.50 | 0.71 |
| High income | reference | | | |
| Age category | | | | |
| 18–30 | reference | | | |
| 31–40 | 1.05 | 0.64 | 0.84 | 1.32 |
| 41–50 | 1.26 | 0.07 | 0.98 | 1.61 |
| 51–64 | 2.10 | <0.01 | 1.65 | 2.67 |
| 65–74 | 8.94 | <0.01 | 3.15 | 25.37 |
| ≥75 | 10.63 | <0.01 | 3.58 | 31.55 |
| Race/Ethnicity | | | | |
| Non-Hispanic White | reference | | | |
| Hispanic | 1.29 | 0.05 | 1.00 | 1.68 |
| Non-Hispanic Black | 1.01 | 0.94 | 0.80 | 1.28 |
| Non-Hispanic Asian | 3.60 | <0.01 | 2.06 | 6.27 |
| Non-Hispanic other race | 1.21 | 0.31 | 0.83 | 1.77 |
| Census region | | | | |
| Northeast | reference | | | |
| Midwest | 0.91 | 0.51 | 0.69 | 1.20 |
| South | 0.69 | <0.01 | 0.56 | 0.86 |
| West | 0.74 | 0.01 | 0.59 | 0.92 |
| Type of health insurance | | | | |
| Private health insurance | reference | | | |
| Medicaid | 0.58 | <0.01 | 0.47 | 0.73 |
| Medicare | 0.45 | <0.01 | 0.35 | 0.57 |
| Uninsured | 0.35 | 0.05 | 0.12 | 0.99 |
| Marital status | | | | |
| Married | reference | | | |
| Widowed | 0.97 | 0.88 | 0.67 | 1.40 |
| Divorced | 0.67 | <0.01 | 0.54 | 0.83 |
| Separated | 0.66 | 0.04 | 0.45 | 0.98 |
| Never married | 0.98 | 0.82 | 0.83 | 1.16 |
| High school graduate | 1.46 | <0.01 | 1.23 | 1.73 |
| USA born | 0.67 | <0.01 | 0.51 | 0.87 |
| Hypertension | 0.99 | 0.91 | 0.84 | 1.16 |
| Diabetes | 1.16 | 0.24 | 0.91 | 1.48 |
| Asthma | 1.31 | 0.01 | 1.07 | 1.61 |
| Any cancer | 1.32 | 0.05 | 1.00 | 1.72 |
| Number of office visits | 1.03 | <0.01 | 1.01 | 1.04 |
| Not English speaking | 0.98 | 0.89 | 0.70 | 1.37 |
| Usual source of care provider | 1.26 | 0.01 | 1.07 | 1.47 |

had a usual source of care provider (aOR = 1.18, p = 0.035) and who had higher office visits (aOR = 1.02, p<0.01) were more likely to receive a COVID-19 vaccine (Table 4).

Table 5 summarizes factors associated with receiving a booster shot among those who had taken the COVID-19 vaccine. A total of 182,843,937 (unweighted n = 16,260) subjects got a booster shot among those who had taken the COVID-19 vaccine. Those who were in the lower income groups compared to those in the high-income group had reduced odds of receiving a booster shot. The poor income group had 33% lower odds of receiving a booster shot (aOR = 0.67, p<0.01), while the low-income group had 42% lower odds of receiving a booster shot (aOR = 0.58, p<0.01).

Table 6 presented differences in male and female vaccination rates across income groups. In the poor income group, females had a slightly lower vaccination rate than males, but this difference was not statistically significant (61.61% vs. 61.93%, p = 0.71). However, females had higher vaccination rates than males in the other income groups. For example, in the middle-income group, females had about a 6% higher vaccination rate than males (74.90% vs. 68.72, p<0.01), and in the high-income group, females had about a 5% higher vaccination rate than males (87.03% vs. 81.85%, p<0.01).

## Discussion

This analysis using MEPS data highlighted disparities in COVID-19 vaccine and booster uptake in the US especially for people in the lower income groups. We observed that the odds of being vaccinated for COVID-19 decreased as income levels decreased, with the poor income group having the least odds of receiving COVID-19 vaccines. Our results show that females who were in the lower income groups could experience more disparity in access to COVID-19 vaccines. Other factors such as race/ethnicity, census region, type of health insurance, number of office visits and marital status were also associated with disparities in COVID-19 vaccine and booster uptake.

Recent guidelines from the Centers for Disease Control and Prevention (CDC) recommends full vaccination as the best way to protect against serious illness from COVID-19 and prevent deaths for everyone, with the added goal of reducing the extra burden the disease is imposing on communities already experiencing disparities [25]. Examining the issue of disparities in COVID-19 vaccine and booster uptake is crucial in achieving this goal. Studies by Nguyen et al. [11] and Williams et al. [26] using the US Census Bureau Household Pulse Survey data from January 2021 to March 2021 showed that COVID-19 vaccination intent differed by race/ethnicity, household income level and age group. Nguyen's findings suggest an interaction between race/ethnicity and household income as a major driver of disparities in COVID-19 vaccination [11]. Our results using a more comprehensive dataset were consistent with the findings of these 2 studies. All other income groups compared the high-income group for all adults in our study had significantly lower odds of being vaccinated for COVID-19 and the poor income group had the worst odds. Our findings provide detailed insight into the role income level play in COVID-19 vaccination disparities because the data used captures vaccination doses as well as booster shots.

When stratified by gender, we observe even worse disparities in COVID-19 vaccination for women in lower income groups than we see for men in similar income groups. This is a noteworthy finding as previous studies have not reported this. A study by Cheng and Li found that compared to Medicare beneficiaries who earned an annual income of $25,000 or more, those who earned less than $25,000 were significantly less likely to be vaccinated [27]. Additionally, other studies have reported that those with lower annual earnings demonstrated less willingness to receive vaccinations [28–30], but none of these studies presented the association of income

**Table 4. Factors associated with taking a COVID-19 vaccine (female only n = 124,060,468).**

| COVID-19 vaccine | Odds ratio | p-value | 95% CI | |
|---|---|---|---|---|
| Family income as % of poverty line | | | | |
| Poor | 0.36 | <0.01 | 0.28 | 0.45 |
| New poor | 0.42 | <0.01 | 0.30 | 0.59 |
| Low income | 0.49 | <0.01 | 0.39 | 0.61 |
| Middle income | 0.51 | <0.01 | 0.43 | 0.62 |
| High income | reference | | | |
| Age category | | | | |
| 18–30 | reference | | | |
| 31–40 | 1.11 | 0.29 | 0.91 | 1.36 |
| 41–50 | 1.41 | 0.01 | 1.10 | 1.79 |
| 51–64 | 2.11 | <0.01 | 1.69 | 2.64 |
| 65–74 | 4.94 | <0.01 | 1.77 | 13.80 |
| ≥75 | 7.25 | <0.01 | 2.44 | 21.55 |
| Race/Ethnicity | | | | |
| Non-Hispanic White | reference | | | |
| Hispanic | 1.70 | <0.01 | 1.33 | 2.16 |
| Non-Hispanic Black | 1.12 | 0.27 | 0.91 | 1.37 |
| Non-Hispanic Asian | 3.06 | <0.01 | 1.81 | 5.19 |
| Non-Hispanic other race | 0.83 | 0.40 | 0.53 | 1.29 |
| Census region | | | | |
| Northeast | reference | | | |
| Midwest | 1.35 | 0.03 | 1.02 | 1.79 |
| South | 0.79 | 0.06 | 0.63 | 1.01 |
| West | 0.80 | 0.05 | 0.64 | 1.00 |
| Type of health insurance | | | | |
| Private health insurance | reference | | | |
| Medicaid | 0.52 | <0.01 | 0.43 | 0.62 |
| Medicare | 0.52 | <0.01 | 0.40 | 0.66 |
| Uninsured | 0.52 | 0.22 | 0.18 | 1.49 |
| Marital status | | | | |
| Married | reference | | | |
| Widowed | 0.81 | 0.12 | 0.62 | 1.05 |
| Divorced | 0.87 | 0.13 | 0.73 | 1.04 |
| Separated | 0.72 | 0.02 | 0.54 | 0.95 |
| Never married | 1.21 | 0.04 | 1.01 | 1.45 |
| High school graduate | 1.41 | <0.01 | 1.18 | 1.70 |
| USA born | 0.72 | 0.01 | 0.56 | 0.91 |
| Hypertension | 1.20 | 0.03 | 1.02 | 1.40 |
| Diabetes | 1.14 | 0.19 | 0.93 | 1.40 |
| Asthma | 1.01 | 0.88 | 0.87 | 1.17 |
| Any cancer | 0.96 | 0.69 | 0.77 | 1.19 |
| Number of office visits | 1.02 | <0.01 | 1.01 | 1.03 |
| Not English speaking | 0.98 | 0.91 | 0.69 | 1.39 |
| Usual source of care provider | 1.18 | 0.04 | 1.01 | 1.38 |

**Table 5. Factors associated with taking a booster shot (those who had a COVID-19 vaccine before, n = 182,843,937).**

| COVID-19 vaccine | Odds ratio | p-value | 95% CI | |
|---|---|---|---|---|
| Family income as % of poverty line | | | | |
| Poor | 0.67 | <0.01 | 0.57 | 0.80 |
| New poor | 0.62 | <0.01 | 0.47 | 0.82 |
| Low income | 0.58 | <0.01 | 0.49 | 0.70 |
| Middle income | 0.72 | <0.01 | 0.63 | 0.82 |
| High income | reference | | | |
| Age category | | | | |
| 18–30 | reference | | | |
| 31–40 | 1.24 | 0.02 | 1.03 | 1.50 |
| 41–50 | 1.26 | 0.02 | 1.04 | 1.52 |
| 51–64 | 1.83 | <0.01 | 1.51 | 2.23 |
| 65–74 | 2.78 | <0.01 | 1.47 | 5.23 |
| ≥75 | 3.34 | <0.01 | 1.77 | 6.29 |
| Female | 1.18 | <0.01 | 1.09 | 1.26 |
| Race/Ethnicity | | | | |
| Non-Hispanic White | reference | | | |
| Hispanic | 1.02 | 0.81 | 0.85 | 1.23 |
| Non-Hispanic Black | 0.89 | 0.18 | 0.75 | 1.06 |
| Non-Hispanic Asian | 1.57 | <0.01 | 1.16 | 2.13 |
| Non-Hispanic other race | 0.84 | 0.34 | 0.58 | 1.21 |
| Census region | | | | |
| Northeast | reference | | | |
| Midwest | 0.99 | 0.92 | 0.79 | 1.24 |
| South | 0.95 | 0.61 | 0.79 | 1.14 |
| West | 0.75 | <0.01 | 0.64 | 0.88 |
| Type of health insurance | | | | |
| Private health insurance | reference | | | |
| Medicaid | 0.77 | 0.01 | 0.63 | 0.93 |
| Medicare | 0.71 | 0.01 | 0.56 | 0.90 |
| Uninsured | 1.29 | 0.41 | 0.70 | 2.39 |
| Marital status | | | | |
| Married | reference | | | |
| Widowed | 0.76 | 0.01 | 0.63 | 0.92 |
| Divorced | 0.72 | <0.01 | 0.62 | 0.84 |
| Separated | 0.52 | <0.01 | 0.37 | 0.72 |
| Never married | 0.93 | 0.31 | 0.82 | 1.07 |
| High school graduate | 1.46 | <0.01 | 1.26 | 1.69 |
| USA born | 1.09 | 0.31 | 0.92 | 1.29 |
| Hypertension | 0.99 | 0.81 | 0.88 | 1.10 |
| Diabetes | 1.09 | 0.19 | 0.96 | 1.25 |
| Asthma | 1.08 | 0.23 | 0.95 | 1.22 |
| Any cancer | 1.17 | 0.05 | 1.00 | 1.38 |
| Number of office visits | 1.01 | <0.01 | 1.01 | 1.02 |
| Not English speaking | 1.00 | 0.98 | 0.77 | 1.29 |
| Usual source of care provider | 1.15 | 0.04 | 1.01 | 1.32 |

**Table 6. Comparison of male and female vaccination rates by income group.**

| | Vaccination rates | | |
|---|---|---|---|
| Family income as % of poverty line (%) | Male | Female | p-value |
| Poor | 61.93 | 61.16 | 0.71 |
| New poor | 66.99 | 67.67 | 0.86 |
| Low income | 67.36 | 72.32 | 0.01 |
| Middle income | 68.72 | 74.90 | <0.01 |
| High income | 81.85 | 87.03 | <0.01 |

and COVID-19 vaccination by gender. Morales et al. found an intersection between gender and socioeconomic status, indicating that women living in poverty were more vaccine hesitant [31]. However, this study used a limited data sample collected between 17 February 2021 and 1 March 2021 when COVID-19 vaccines were not yet available to all age groups. Our findings align with that of Morales et al.'s study. We leveraged data from a time when the vaccine was available and recommended for all age group, thereby strengthening the quality of evidence. There could be several reasons for observing worse COVID-19 vaccination disparities for women in lower income groups. It is known that the American healthcare system has long been plagued with gender disparities in health and utilization of healthcare. These disparities have been observed in outpatient care, surgery, preventative services, inpatient hospital services, and utilization of physicians and home care [32, 33]. Black women are disproportionately affected by these disparities as they have the worst health outcomes and shorter life expectancies compared to women of other races and report being treated poorly when seeking healthcare [34, 35]. Historically, women have been known to have less economic resources than men, and even when they have equal resources, they could still experience more disparities in healthcare [36]. Misinformation and disinformation regarding mRNA vaccines, along with concerns over possible side effects, especially among women of childbearing age, such as concerns about fertility, could have contributed to the observed disparities among women [37].

Despite COVD-19 vaccines and booster being made available free of charge to everyone in the US, we still found lower odds of being vaccinated for subjects who were covered by Medicaid, Medicare or with no insurance compared to those who were covered by private insurance. Earlier studies have attributed this to the plethora of misinformation that surrounded the rollout of COVID-19 vaccines in the US [11], increased financial hardship lower-income families faced due of the pandemic, and the indirect cost of traveling to locate a clinic to receive the vaccine doses [4]. People without insurance are also less likely to have a primary care provider who would advise them and provide necessary information about taking COVID-19 vaccines and booster [38]. Our study showed that women with a usual care provider and more frequent office visits were more likely to receive the COVID-19 vaccine. This underscores the importance of ensuring access to care for national preparedness in facing future pandemics.

With the updated COVID-19 vaccine being rolled out in the US as of September 2023, disparities by income levels and insurance type could worsen as the government no longer covers the cost of purchasing and distributing the shots. There has been reported cases of difficulty accessing the updated vaccine and confusion over insurance coverage for many Americans as some people who have tried to get the vaccine encountered unexpected bills or cancelled appointments [39]. People who fall in lower-income groups or those who do not have insurance coverage may be disproportionately affected as a result of this.

Disparities in COVID-19 vaccine uptake has several implications for the socially vulnerable and the public health of the US as a whole. According to a forecast by the Economist

Intelligence Unit (EIU), COVID-19 vaccine delays and inequity could cost the global economy as much as US $2.3 trillion, with developing countries shouldering about two-thirds of these losses [40]. The US could bear some of these losses as a result of lost productivity, morbidity and mortality. People in the lower income groups would likely be most affected as they are known to be disproportionately affected by COVID-19. These disparities also worsen lack of trust and health inequities which has existed even before the pandemic.

There are some limitations to this study that could impact our study findings. First, because our results were based on the adult population ($\geq$18 years old in 2021), they may not be generalizable to adolescents or children. Second, emergency visits and hospital admissions that could potentially be associated with the outcomes were not counted in the total number of visits. Third, the self-reported nature of COVID-19 vaccine and booster uptake as well as disease diagnosis may not accurately reflect the actual vaccination status or history of every study subject. Finally, because this study was based on a measured survey dataset, unmeasured confounders could have an impact on our findings.

## Conclusions

Our study shows that being in a lower income group, among other factors, was associated with lower odds of COVID-19 vaccine and booster uptake, and females with lower income had even greater disparities in COVID-19 vaccine access than males with similar income. There is need to build on efforts already put in place by federal and state governments to curb these disparities such as removing financial barriers to access, community outreaches and mobile/pop-up vaccine clinics. Sustained financial covering especially for those in lower income groups and those without health insurance is needed as the updated vaccines are rolled out in the US. Our findings can guide policy efforts to equitably distribute vaccines and stop COVID-19 health inequities from worsening.

## Author Contributions

**Conceptualization:** Kenechukwu C. Ben-Umeh, Jaewhan Kim.

**Data curation:** Kenechukwu C. Ben-Umeh, Jaewhan Kim.

**Formal analysis:** Jaewhan Kim.

**Investigation:** Kenechukwu C. Ben-Umeh, Jaewhan Kim.

**Methodology:** Jaewhan Kim.

**Writing – original draft:** Kenechukwu C. Ben-Umeh, Jaewhan Kim.

**Writing – review & editing:** Kenechukwu C. Ben-Umeh, Jaewhan Kim.

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
