## [Decision Letter · Decision Letter 0]

2 Jan 2024

PONE-D-23-37178Income disparities in COVID-19 vaccine and booster uptake in the United States: An analysis of cross-sectional data from the Medical Expenditure Panel SurveyPLOS ONE

Dear Dr. Kim,

Thank you for submitting your manuscript to PLOS ONE. After careful consideration, we feel that it has merit but does not fully meet PLOS ONE’s publication criteria as it currently stands. Therefore, we invite you to submit a revised version of the manuscript that addresses the points raised during the review process.

We look forward to receiving your revised manuscript.

Kind regards,

Benjamin Ansa

Academic Editor

PLOS ONE

Journal Requirements:

Additional Editor Comments:

This is a nicely written manuscript of a study that explored the disparities in COVID-19 vaccine and booster uptake by income levels. There are few minor comments for author consideration in order to improve the quality of the paper.

1. In the methodology, include the response rates and reliability/validity of the survey instruments for the year analyzed. Also, clarify what is meant by "office visit".

2. Read carefully the comments of the three other reviewers and respond to them appropriately during the revision process.

Reviewers' comments:

Reviewer's Responses to Questions

**Comments to the Author**

1. Is the manuscript technically sound, and do the data support the conclusions?

Reviewer #1: Yes

Reviewer #2: Yes

Reviewer #3: Yes

2. Has the statistical analysis been performed appropriately and rigorously? 

Reviewer #1: Yes

Reviewer #2: Yes

Reviewer #3: Yes

3. Have the authors made all data underlying the findings in their manuscript fully available?

Reviewer #1: Yes

Reviewer #2: Yes

Reviewer #3: Yes

4. Is the manuscript presented in an intelligible fashion and written in standard English?

Reviewer #1: Yes

Reviewer #2: Yes

Reviewer #3: Yes

5. Review Comments to the Author:

Reviewer #1: As a reviewer for the manuscript titled "Income Disparities in COVID-19 Vaccine and booster uptake in the United States: An Analysis of Cross-sectional Data from the Medical Expenditure Panel Survey," I have evaluated it based on the provided criteria. Here are my comments:

1. Technical Soundness and Data Support for Conclusions:

• The manuscript presents a technically sound piece of research. It utilises data from the Medical Expenditure Panel Survey (MEPS) 2021, a nationally representative sample. The study employs weighted logistic regression to analyse factors associated with COVID-19 vaccine and booster uptake. The methodology follows the Strengthening the Reporting of Observational Studies in Epidemiology (STROBE) reporting guideline, which adds to the technical robustness of the study.

• The data analysis reveals significant disparities in COVID-19 vaccine and booster uptake among different income groups in the United States. For instance, individuals in the poor-income group were 55% less likely to be vaccinated than those in the high-income group. This supports the study's conclusions about income-related disparities in vaccination rates.

2. Statistical Analysis:

• The statistical analysis appears to have been conducted rigorously and appropriately. The study uses weighted logistic regression, and the Taylor-series linearisation method is employed to calculate correct standard errors, reflecting the complex sample design. P-values less than 0.05 were considered statistically significant, a standard practice in such analyses.

3. Data Availability:

• The manuscript adheres to the PLOS Data policy as it makes use of publicly available data from the MEPS 2021 Full Year Consolidated File. This ensures compliance with the requirement for completing all data underlying the findings fully functional without restriction.

4. Presentation and Language:

• The manuscript is presented in an intelligible fashion and is written in standard English. All sections are clear and concise, effectively communicating the study's objectives, methodology, and findings.

In conclusion, the manuscript is well-constructed, with a sound methodology, rigorous statistical analysis, adherence to data availability policies, and clear presentation. The findings contribute valuable insights into the disparities in COVID-19 vaccine and booster uptake across different income levels in the United States.

Reviewer #2: From my reading, this manuscript describes a study in which the authors used publicly available survey data to assess the impact of income on COVID-19 vaccine uptake. The authors show that lower income is associated with a lower likelihood of COVID-19 vaccination. They also show that this disparity in vaccine uptake by income group is more drastic for females than males. Similarly, the authors show that uninsured individuals and those insured by Medicare or Medicaid have lower rates of vaccine uptake compared to those insured by private companies. Overall, this manuscript is written clearly and uses simple statistical analyses to effectively support the authors' ultimate conclusions.

The only additional statistical analysis I would suggest is a statistical comparison of male and female vaccination rates by income group to show whether or not the more drastic disparities seen for females are statistically significant. Beyond that, my only other recommendation would be to edit once or twice more for grammatical inconsistencies or errors such as "less likelihood" rather than "a lower likelihood."

Reviewer #3: Authors aimed to evaluate covid-19 vaccine and booster uptake by income levels using a nationally representative adult (≥18 years old) sample from the 2021 Medical Expenditure Panel Survey (MEPS). Authors identified inequities in vaccine uptake according to income with individuals with higher income level with higher odds of being vaccinated.

The current study provides some helpful information, however, there are some issues as currently presented.

1) In the abstract, are regression adjusted? If they are, then authors could make this clearer for the reader by noting aOR instead of OR

2)Authors should be consistent throughout the manuscript as they use “likelihood” when they might be referring to the odds. Here is an example in line 149:

“Those who were in the lower income groups compared to those in the high-income group had lower likelihood of getting a booster shot.”

3)Line 142 of result:

“These numbers might indicate that women with lower incomes could experience more disparity in access to COVID-19 vaccines even more than men with lower incomes.”

In the discussion, it would be helpful if authors expand on potential reasons for this observation. For example, misinformation regarding mRNA vaccine might have explained why women of childbearing age might have had lower vaccine uptake when compared to men.

4)Line 223:

Authors note the following:

“Second, both office visits and outpatient visits were used to calculate the overall number of healthcare visits”

What is the difference between office visit and outpatient visit? Could some classification have led to duplicated entries?

5) line 145

Authors could consider underscoring in the discussion, the public health implications of the following observation:

“Women who had a usual source of care provider (OR=1.18, p=0.035) and who had higher office visits (OR=1.02, p<0.01) were more likely to receive a COVID-19 vaccine (Table 4).”

For example, providing access to care will be an important step to ensure that the nation is ready for future pandemics.

Overall, this is a helpful analysis that complements and extends information already available in the literature about the association between income and lower vaccine and booster uptake; however, the central finding of inequities according to income level is not particularly novel.

6. PLOS authors have the option to publish the peer review history of their article (what does this mean?). If published, this will include your full peer review and any attached files.

Reviewer #1: **Yes: **Harry James Gaffney

Reviewer #2: No

Reviewer #3: No

---

## [Author Response · Author response to Decision Letter 0]

22 Jan 2024

January 19, 2024

Dear Dr. Ansa and Reviewers, 

We appreciate the opportunity to revise our submitted manuscript, Income disparities in COVID-19 vaccine and booster uptake in the United States: An analysis of cross-sectional data from the Medical Expenditure Panel Survey. Comments from the reviewers were very helpful in improving the article's focus and we believe the suggestions have made this paper much stronger. Below are the specific reviewer’s comments (in bold) followed by our responses. Newly added parts in the manuscript are highlighted in yellow. Thank you again for taking the time to consider our manuscript. 

Journal Requirements:

Response: We double-checked the style requirements, and everything is in good shape. 

Response: We reviewed the reference list, and everything appears to be in order. 

Additional Editor Comments:

This is a nicely written manuscript of a study that explored the disparities in COVID-19 vaccine and booster uptake by income levels. There are few minor comments for author consideration in order to improve the quality of the paper.

1. In the methodology, include the response rates and reliability/validity of the survey instruments for the year analyzed. 

Response: Thank you for the comments. The response rates in the 2021 survey varied based on the four data collection panels, with a range between 60% and 87.6%. We have added this information under the Data section in the manuscript as follows:

In the 2021 survey, the response rates varied between 60% to 87.6% based on the four data collection panels (referred to as Panel 23 to Panel 26). [19]

Since its initiation in 1996, the MEPS has been a valuable resource for researchers, contributing to numerous publications. The reliability and validity of the MEPS data have been assessed in multiple publications. We have included this information in the Data section of the manuscript, highlighting the reliability and validity established through prior research: 

The MEPS has been conducted since 1996, and numerous research studies have utilized this data; hence its reliability and validity have been established by previous publications. [20, 21, 22]

References 

19. Agency for Healthcare Research and Quality. MEPS Annual Methodology Report 2021 [Accessed January 13 2024]. Available from: https://meps.ahrq.gov/data_files/publications/annual_contractor_report/MEPS-Methodology-Report-2021.html#Tfour4

20. Zuvekas S. Validity of Household Reports of Medicare-covered Home Health Agency Use. Agency for Healthcare Research and Quality Working Paper No. 09003, August 2009, https://meps.ahrq.gov/data_files/publications/workingpapers/wp_09003.pdf

21. Shah CH, Brown JD. Reliability and Validity of the Short-Form 12 Item Version 2 (SF-12v2) Health-Related Quality of Life Survey and Disutilities Associated with Relevant Conditions in the U.S. Older Adult Population. J Clin Med. 2020 Feb 29;9(3):661. doi: 10.3390/jcm9030661. PMID: 32121371; PMCID: PMC7141358.

22. Hayes CJ, Bhandari NR, Kathe N, Payakachat N. Reliability and Validity of the Medical Outcomes Study Short Form-12 Version 2 (SF-12v2) in Adults with Non-Cancer Pain. Healthcare (Basel). 2017 Apr 26;5(2):22. doi: 10.3390/healthcare5020022. PMID: 28445438; PMCID: PMC5492025.

1-1. Also, clarify what is meant by "office visit".

The MEPS website (https://meps.ahrq.gov/mepsweb/data_stats/MEPS_topics.jsp?topicid=36Z-1) provides the definition of office visit as follows:

“Office-based visits/use/events can occur in a variety of places such as a doctor's or group practice office, medical clinic, managed care plan or hmo center, neighborhood/family/community health center, surgical center, rural health clinic, company clinic, school clinic, walk-in urgent centers, VA facility, or laboratory/x-ray facilities.”

We added the following sentence with a new reference in the Covariates section to define office visit in the manuscript:

Office visit was defined as care provided in a doctor’s office, group practice office, medical clinic, community health center, urgent care center, or school clinic [23]. 

Reference

23. Agency for Healthcare Research and Quality. MEPS Topics: Office-Based Visits/Use/Events and Expenditures [Accessed January 13 2024]. Available from: https://meps.ahrq.gov/mepsweb/data_stats/MEPS_topics.jsp?topicid=36Z-1

2. Read carefully the comments of the three other reviewers and respond to them appropriately during the revision process.

Response: Thank you. We thoroughly read the comments and made a dedicated effort to address each one. 

Reviewers' comments:

Review Comments to the Author:

Reviewer #1: As a reviewer for the manuscript titled "Income Disparities in COVID-19 Vaccine and booster uptake in the United States: An Analysis of Cross-sectional Data from the Medical Expenditure Panel Survey," I have evaluated it based on the provided criteria. Here are my comments:

1. Technical Soundness and Data Support for Conclusions:

• The manuscript presents a technically sound piece of research. It utilises data from the Medical Expenditure Panel Survey (MEPS) 2021, a nationally representative sample. The study employs weighted logistic regression to analyse factors associated with COVID-19 vaccine and booster uptake. The methodology follows the Strengthening the Reporting of Observational Studies in Epidemiology (STROBE) reporting guideline, which adds to the technical robustness of the study.

• The data analysis reveals significant disparities in COVID-19 vaccine and booster uptake among different income groups in the United States. For instance, individuals in the poor-income group were 55% less likely to be vaccinated than those in the high-income group. This supports the study's conclusions about income-related disparities in vaccination rates.

Response: Thank you for the evaluation!

2. Statistical Analysis:

• The statistical analysis appears to have been conducted rigorously and appropriately. The study uses weighted logistic regression, and the Taylor-series linearisation method is employed to calculate correct standard errors, reflecting the complex sample design. P-values less than 0.05 were considered statistically significant, a standard practice in such analyses.

Response: Thank you. 

3. Data Availability:

• The manuscript adheres to the PLOS Data policy as it makes use of publicly available data from the MEPS 2021 Full Year Consolidated File. This ensures compliance with the requirement for completing all data underlying the findings fully functional without restriction.

Response: Thank you!

4. Presentation and Language:

• The manuscript is presented in an intelligible fashion and is written in standard English. All sections are clear and concise, effectively communicating the study's objectives, methodology, and findings.

Response: Thank you for the comment. 

In conclusion, the manuscript is well-constructed, with a sound methodology, rigorous statistical analysis, adherence to data availability policies, and clear presentation. The findings contribute valuable insights into the disparities in COVID-19 vaccine and booster uptake across different income levels in the United States.

Response: We appreciate the time you took to review the manuscript. 

Reviewer #2: From my reading, this manuscript describes a study in which the authors used publicly available survey data to assess the impact of income on COVID-19 vaccine uptake. The authors show that lower income is associated with a lower likelihood of COVID-19 vaccination. They also show that this disparity in vaccine uptake by income group is more drastic for females than males. Similarly, the authors show that uninsured individuals and those insured by Medicare or Medicaid have lower rates of vaccine uptake compared to those insured by private companies. Overall, this manuscript is written clearly and uses simple statistical analyses to effectively support the authors' ultimate conclusions.

1. The only additional statistical analysis I would suggest is a statistical comparison of male and female vaccination rates by income group to show whether or not the more drastic disparities seen for females are statistically significant. 

Response: Thank you for the comment. Table 2 through Table 5 indicated that females had higher vaccination rates than males after controlling for potential confounders. However, upon examining vaccination rates exclusively for males (Table 3) or females (Table 4) within income groups, there could be potentially more significant disparities in vaccination rates among females than males. 

Since we did not initially report the vaccination rates by income group for both males and females, we added Table 6 in the Results section. This table presents a comparison of male and female vaccination rates by income group. With the exception of the poor income group, females in the other income groups had higher vaccination rates than males. 

The following descriptions, along with Table 6, have been added to the manuscript in the Results section: 

Table 6 presented differences in male and female vaccination rates across income groups. In the poor income group, females had a slightly lower vaccination rate than males, but this difference was not statistically significant (61.61% vs. 61.93%, p=0.71). However, females had higher vaccination rates than males in the other income groups. For example, in the middle-income group, females had about a 6% higher vaccination rate than males (74.90% vs. 68.72, p<0.01), and in the high-income group, females had about a 5% higher vaccination rate than males (87.03% vs. 81.85%, p<0.01). 

Table 6. Comparison of male and female vaccination rates by income group

 Vaccination rates 

Family income as % of poverty line (%) Male Female p-value 

 Poor 61.93 61.16 0.71

 New poor 66.99 67.67 0.86

 Low income 67.36 72.32 0.01

 Middle income 68.72 74.90 <0.01

 High income 81.85 87.03 <0.01

2. Beyond that, my only other recommendation would be to edit once or twice more for grammatical inconsistencies or errors such as "less likelihood" rather than "a lower likelihood."

Response: Thank you for the recommendation. Since the results were based on odds ratios from logistic regression, we carefully reviewed the manuscript, replacing occurrences of “likelihood” with “odds” in the Abstract, Results, and Discussion sections.

Reviewer #3: Authors aimed to evaluate covid-19 vaccine and booster uptake by income levels using a nationally representative adult (≥18 years old) sample from the 2021 Medical Expenditure Panel Survey (MEPS). Authors identified inequities in vaccine uptake according to income with individuals with higher income level with higher odds of being vaccinated.

The current study provides some helpful information, however, there are some issues as currently presented.

1) In the abstract, are regression adjusted? If they are, then authors could make this clearer for the reader by noting aOR instead of OR

Response: Thank you for the suggestion. All regression results were adjusted. In the Abstract and the Results sections, we replaced OR with aOR to indicate adjusted odds ratio. 

2)Authors should be consistent throughout the manuscript as they use “likelihood” when they might be referring to the odds. Here is an example in line 149:

“Those who were in the lower income groups compared to those in the high-income group had lower likelihood of getting a booster shot.”

Response: Thank you for the comment. We agree with the need for consistency. Since the results were based on odds ratios from logistic regression, we carefully reviewed the manuscript, replacing occurrences of “likelihood” with “odds” in the Abstract, Results, and Discussion sections. 

3)Line 142 of result:

“These numbers might indicate that women with lower incomes could experience more disparity in access to COVID-19 vaccines even more than men with lower incomes.”

In the discussion, it would be helpful if authors expand on potential reasons for this observation. For example, misinformation regarding mRNA vaccine might have explained why women of childbearing age might have had lower vaccine uptake when compared to men.

Response: Thank you for the great suggestion. We added the following sentence with a new reference in the Discussion section: 

Misinformation and disinformation regarding mRNA vaccines, along with concerns over possible side effects, especially among women of childbearing age, such as concerns about fertility, could have contributed to the observed disparities among women [37]. 

Reference 

37. Nassiri-Ansari T, Atuhebwe P, Ayisi AS, Goulding S, Johri M, Allotey P, Schwalbe N. Shifting gender barriers in immunisation in the COVID-19 pandemic response and beyond. Lancet. 2022;400(10345):24.

4)Line 223:

Authors note the following:

“Second, both office visits and outpatient visits were used to calculate the overall number of healthcare visits”

What is the difference between office visit and outpatient visit? Could some classification have led to duplicated entries?

Response: Thank you for the comment. We double-checked the definitions of the office visit and the outpatient visit from the MEPS website. Below are the definitions:

“Office-based visits/use/events can occur in a variety of places such as a doctor's or group practice office, medical clinic, managed care plan or hmo center, neighborhood/family/community health center, surgical center, rural health clinic, company clinic, school clinic, walk-in urgent centers, VA facility, or laboratory/x-ray facilities.”

“An outpatient department visit/use/event is any visit made during the person's reference period to a hospital outpatient department, such as a unit of a hospital, or a facility connected with a hospital, providing health and medical services to individuals who receive services from the hospital but do not require hospitalization overnight.”

According to their definitions, it is less likely to have double counting between the two visits. Also, as we did not control the number of outpatient visits in the regression, this should not be considered a limitation. Thus, we removed this limitation from the Discussion section:

“both office visits and outpatient visits were used to calculate the overall number of healthcare visits. It is possible that duplicate visits were recorded.”

References 

Agency for Healthcare Research and Quality. MEPS Topics: Office-Based Visits/Use/Events and Expenditures. Available from: https://meps.ahrq.gov/mepsweb/data_stats/MEPS_topics.jsp?topicid=36Z-1

Agency for Healthcare Research and Quality. MEPS Topics: Outpatient Visits/Use/Events and Expenditures. Available from: https://meps.ahrq.gov/mepsweb/data_stats/MEPS_topics.jsp?topicid=38Z-1

5) line 145

Authors could consider underscoring in the discussion, the public health implications of the following observation:

“Women who had a usual source of care provider (OR=1.18, p=0.035) and who had higher office visits (OR=1.02, p<0.01) were more likely to receive a COVID-19 vaccine (Table 4).”

For example, providing access to care will be an important step to ensure that the nation is ready for future pandemics.

Response: Thank you for the suggestion. We added the following sentences under the Discussion section:

Our study showed that women with a usual care provider and more frequent office visits were more likely to receive the COVID-19 vaccine. This underscores the importance of ensuring access to care for national preparedness in facing future pandemics.

Overall, this is a helpful analysis that complements and extends information already available in the literature about the association between income and lower vaccine and booster uptake; however, the central finding of inequities according to income level is not particularly novel.

Thank you very much again for your time and thoughtful comments.

---

## [Editor Report · Decision Letter 1]

31 Jan 2024

Income disparities in COVID-19 vaccine and booster uptake in the United States: An analysis of cross-sectional data from the Medical Expenditure Panel Survey

PONE-D-23-37178R1

Dear Dr. Jaewhan Kim,

We’re pleased to inform you that your manuscript has been judged scientifically suitable for publication and will be formally accepted for publication once it meets all outstanding technical requirements.

Kind regards,

Benjamin Ansa

Academic Editor

PLOS ONE

Additional Editor Comments (optional):

The authors have responded adequately to all of the reviewers' comments and the quality of the manuscript has been remarkably improved by the minor revision.
---

## [Editor Report · Acceptance letter]

12 Feb 2024

PONE-D-23-37178R1 

PLOS ONE

Dear Dr. Kim, 

I'm pleased to inform you that your manuscript has been deemed suitable for publication in PLOS ONE. Congratulations! Your manuscript is now being handed over to our production team.

Kind regards, 

on behalf of

Dr. Benjamin Ansa 

Academic Editor

PLOS ONE